# Vulnerable Narcissism and Problematic Social Networking Sites Use: Focusing the Lens on Specific Motivations for Social Networking Sites Use

**DOI:** 10.3390/healthcare10091719

**Published:** 2022-09-08

**Authors:** Alessandro Musetti, Valentina Grazia, Alessia Alessandra, Christian Franceschini, Paola Corsano, Claudia Marino

**Affiliations:** 1Department of Humanities, Social Sciences and Cultural Industries, University of Parma, 43121 Parma, Italy; 2Department of Medicine and Surgery, University of Parma, 43125 Parma, Italy; 3Department of Developmental and Social Psychology, University of Padova, 35131 Padova, Italy

**Keywords:** vulnerable narcissism, motives, problematic social networking sites use, addictive behaviors

## Abstract

Research highlighted that Problematic Social Networking Sites Use (PSNSU) and vulnerable narcissism are associated. However, the mechanisms underlying this relationship are still unclear. The present study aimed to test the mediating role of motives for social networking sites (SNSs) use between vulnerable narcissism and five symptoms of PSNSU (i.e., preference for online social interactions, mood regulation, cognitive preoccupation, compulsive use, and negative outcomes) in a sole model. Self-report questionnaires were completed by 344 SNSs users in the age range of 18–30 years (76.5% females; mean age = 23.80 years, standard deviation = 2.30 years). Vulnerable narcissism, three motives to use SNSs (coping, conformity, enhancement), and symptoms of PSNSU were assessed. Structural equation modeling was used to test for mediation. The results indicate that both motives with positive (i.e., enhancement) and negative (i.e., coping and conformity) valence partially mediated the association between vulnerable narcissism and different symptoms of PSNSU. We conclude that individuals with vulnerable narcissism may develop PSNSU not only as a compensatory strategy to cope with psychosocial difficulties but also as a result of a gratification-seeking process.

## 1. Introduction

Social networking sites (SNSs) have become part of youth’s everyday life. Over 3.6 billion people were using SNSs worldwide in 2020, and the number of SNSs users has increased substantially in the past decade [1]. Although most people benefit from using SNSs (see [2] for a review), a minority of individuals use these platforms excessively and uncontrollably ([3]). According to Andreassen and Pallesen [4], Problematic Social Networking Sites Use (PSNSU) is characterized by: (1) excessive concern about SNSs or online-communication applications, (2) strong motivation to log on to or to use SNSs, and (3) time and effort spent on SNSs use impair other social activities, studies/work, interpersonal relationships and/or psychological health and well-being. This conceptualization relies on the six-component (i.e., salience, mood modification, tolerance, withdrawal, conflict, and relapse) model of addiction developed by Griffiths [5]. According to this perspective, PSNSU can be classed as a “genuine addiction” based on the endorsement of criteria derived from other addictive behaviors [6]. In contrast, other scholars (e.g., [7]) suggest that specific problematic Internet use, such as PSNSU, may be described as secondary manifestations of pre-existing psychopathology (e.g., anxious or depressive symptoms) rather than a primary disorder. Within this perspective, Marino et al. [8] adapted Caplan’s [9,10] Generalized Problematic Internet Use (GPIU) model in the context of SNSs use. In an attempt to overcome the limitations of atheoretical and confirmatory approaches to behavioral addiction (see [11]), the authors identified five key symptoms that are specifically implied in PSNSU: (a) preference for online social interactions (POSI; i.e., believing to be safer, more comfortable and more confident with SNSs interactions than offline); (b) the usage of SNSs for mood regulation (i.e., the use of SNSs to reduce negative, unpleasant feelings, such as anxiety and loneliness); (c) cognitive preoccupation (i.e., obsessive thought patterns about using SNSs); (d) compulsive use of SNSs (i.e., the inability to control time spent on SNSs); (e) negative outcomes of SNSs use (i.e., personal and social impairments due to SNSs use). In addition, the GPIU model considers cognitive preoccupation and compulsive use symptoms as two facets of the more general deficient self-regulation of Internet users. Recently, Svicher et al. [12] adopted a network approach to assess which symptoms are most central to PSNSU in a sample of 1344 young adults. The findings revealed the pivotal role of self-regulation and POSI among symptoms of PSNSU, suggesting both clinical similarities and differences with well-established addictions. 

Research has identified a variety of risk factors for PSNSU, including individual (e.g., attachment anxiety, emotional dysregulation, low self-esteem), interpersonal (e.g., low relationship satisfaction, peer alienation, fear of missing out), and SNS-related (e.g., likes-seeking behaviors, meeting psychological needs through SNSs use, escapism motivation for SNSs use) factors [13]. Over the last decades, a growing body of research has investigated the relationship between PSNSU and narcissism [14] and between PSNSU and motives for SNSs use [15]. However, to the authors’ knowledge, no attempt has been made to explicitly test the association between vulnerable narcissism and PSNSU as mediated by theory-driven motives for using SNSs.

### 1.1. Vulnerable Narcissism and Problematic Social Networking Sites Use

Narcissism is a broad and widely studied personality construct that includes an exaggerated sense of self-importance, need for admiration, entitlement, and lack of empathy [16]. The varieties of narcissistic expressions range along a continuum from normal personality traits to a frank psychiatric disorder (i.e., the narcissistic personality disorder; [17]). Due to their personality features, narcissistic individuals may be prone to engage in SNSs to gain the attention of others and reinforce their grandiose self-concept [18]. Moreover, some characteristics of SNSs, such as asynchronous communication and indirect feedback, allow users some degree of control over their self-presentation and communication with others [19], resulting in a sort of “narcissistic dream” (p. 308) [20].

Starting from the seminal work by Wink (1991), a huge amount of literature has highlighted the dual nature of narcissism in terms of its overt (i.e., grandiose) and covert (i.e., vulnerable) forms [21]. While grandiose narcissism is characterized primarily by excessive self-confidence and manipulative and exploitative behavior, vulnerable narcissism is more likely to manifest in shyness and hypersensitivity to the evaluation of others [22]. Casale and Banchi [14] systematically reviewed 21 studies that examined the relationship between narcissism and PSNSU and found that grandiose narcissism is consistently and positively associated with problematic Facebook use and not consistently associated across studies with PSNSU. As the authors noted, differently from other SNSs (e.g., Twitter), Facebook may be particularly appealing to grandiose narcissists because it is a readily available tool for self-promotions where users can easily share their updates and obtain frequent positive feedback [23]. In addition, [14] found that vulnerable narcissism is consistently positively related to PSNSU, although studies on this topic are still scarce. For example, [24] adopted the GPIU model to assess differences in PSNSU symptoms between grandiose narcissists, vulnerable narcissists, and non-narcissists. The results showed that vulnerable narcissists reported higher levels of all PSNSU symptoms than non-narcissists and higher POSI and global PSNSU levels than grandiose narcissists. Compared to the latter, vulnerable narcissists are characterized by avoidance, social inhibition, and high negative affect, which makes it harder for these individuals to fulfill their narcissistic needs through offline social interactions [25]. Thus, they may be excessively involved in any SNSs to gain a higher level of control over communication with others while obtaining immediate gratification through likes and comments [26]. Building on this evidence, the relationship between vulnerable narcissism and PSNSU may be conceptualized within the theory of compensatory Internet use [27]. According to this theoretical framework, vulnerable narcissists are prone to use SNSs to compensate for social needs due to the lack of offline social relationships and deficiencies in self-esteem, resulting in an increased risk of PSNSU [28]. 

### 1.2. The Mediating Role of Motives for Social Networking Sites Use

Previous studies have outlined a variety of motivations in the attempt to explain why people engage in SNSs use (e.g., socializing, self-presentation, and acquiring information; [19,29]). Beyond motivations related to frequent but nonproblematic SNSs use, a growing body of research has more closely investigated the specific motivations that are more strictly involved in the development of PSNSU (e.g., [30,31]). 

Marino et al. [32] adapted the traditional motivation model for addictive behaviors [33,34] in the context of SNSs use. This model states that people are driven to engage in problematic online behaviors in order to achieve predetermined desired goals. Accordingly, motives for PSNSU have been classified into four categories based on the *valence* (positive or negative) and the *source* (external or internal) of the expected affective change (see Table 1). 

Marino et al. [32] found that motives with negative valences, such as coping (i.e., using SNS to reduce negative affect) and conformity (i.e., using SNS to avoid social rejection), were more closely related to problematic Facebook use than motives with positive valences, such as enhancement (i.e., using SNS to increase positive affect) and social (i.e., using SNS to improve relationships with friends). Consistent with the theory of compensatory Internet use [27], a vicious cycle may start with the person spending time on SNSs to temporarily escape from unpleasant feelings or events related to offline problems, and this, in turn, may exacerbate the sense of being unable to cope with daily difficulties [35]. In addition, findings of a recent study by Balcerowska and Sawicki [28] on 1659 SNSs users showed that compensatory mechanism is specific for narcissistic vulnerability, whereas other forms of narcissism may be linked with PSNSU in other ways (i.e., via antagonism and hostility towards others in the case of rivalrous narcissism). However, the authors also highlighted the need for further empirical studies focusing on the role of users’ motives for SNSs use in the relationship between vulnerable narcissism and PSNSU.

### 1.3. Current Study

Research highlighted that PSNSU and vulnerable narcissism are associated. However, the mechanisms underlying this relationship are still unclear. The present study sought to test a single model in which it is hypothesized that vulnerable narcissism is directly associated with PSNSU symptoms (i.e., POSI, mood regulation, cognitive preoccupation, compulsive behavior, and negative outcomes) [14] and indirectly via different motives for SNSs use (i.e., coping, conformity, enhancement, social) [28]. This study’s aim was twofold. First, based on the GPIU model [8], which considers PSNSU as a multidimensional phenomenon, we aimed to extend previous findings by identifying unique associations between vulnerable narcissism and specific PSNSU symptoms beyond the established association with overall scores of PSNSU. Specifically, based on the previous work by Casale et al. [24], we expected to find positive and direct associations between vulnerable narcissism and all symptoms of PSNSU (H1). Second, we aimed to provide empirical evidence regarding the association of vulnerable narcissism and PSNSU via motives for SNSs use with negative and positive valence [28]. Thus, we hypothesized that coping and conformity motives would mediate the relationship between vulnerable narcissism and PSNSU symptoms and that social and enhancement motives would mediate such association, with motives with negative valence showing stronger effects than motives with positive valence (H2). 

## 2. Method

### 2.1. Participants and Procedure

This cross-sectional study adopted a snowball sampling strategy. An online survey was used to collect data from 1 to 30 April 2021 by means of advertisements shared in social network groups. Before starting the survey, all participants received information about the research goals and scopes. Participation was entirely voluntary, with complete confidentiality and anonymity guaranteed as no personal data or Internet Protocol address was collected. The participants could withdraw from the study at any time. Inclusion criteria were as follows: (i) being between 18 and 30 years of age; (ii) being able to complete questionnaires in Italian; and (iii) using at least one SNS. Participants provided written online consent and completed all measures using Google Forms. In this study, participants were 344 young adults between the ages of 18 and 30 (*M*_age_ = 23.80 years; *SD*_age_ = 2.30 years). Of them, 76.5 % were female, and 99.4% were not married at the time of data collection. 

The study was designed and carried out according to the Ethical Code of the Italian Association of Psychology (AIP), the European Code of Conduct for Research Integrity (ECCRI), and the 1964 Helsinki Declaration and its later amendments. 

### 2.2. Measures

#### 2.2.1. Socio-Demographics

Socio-demographic data included question on age, gender, and family status.

#### 2.2.2. Problematic Social Networking Sites Use

An adaptation of the Problematic Facebook Use Scale [8] was used to measure PSNSU, where we replaced the term “Facebook” with “social networking sites” in all questions [36]. It is a self-report questionnaire including 15 items rated on an 8-point scale from 1 (*definitely disagree*) to 8 (*definitely agree*). The total score is separated into five subscales, each containing 3 items, reflecting symptoms of PSNSU: POSI (e.g., “I prefer online social interactions over face-to-face communication”; Cronbach alpha = 0.74), mood regulation (“I have used social networking sites to make myself feel better when I was down”; Cronbach alpha = 0.76), cognitive preoccupation (e.g., “I think obsessively about going on social networking sites when I am offline”; Cronbach alpha = 0.67), compulsive use (e.g., “I have difficulty controlling the amount of time I spend on social networking sites”; Cronbach alpha = 0.84), negative outcomes (e.g., “My social networking sites use has created problems for me in my life”; Cronbach alpha = 0.76). Higher scores indicate greater PSNSU symptoms. 

#### 2.2.3. Motives for Social Networking Sites Use

We used an adapted version of the Facebook Motives Questionnaire [32,33] to assess users’ motives for SNSs use. Specifically, in each item, the word “Facebook” was replaced with “social networking sites”. Participants rated how often they logged on SNSs with different motivations during the last 12 months. The questionnaire evaluates four motives: coping (e.g., “To forget your worries?”), conformity (e.g., “To be liked by others?”), enhancement (e.g., “Because it is exciting?”), and social motive (e.g., “To come into contact with others?”). The scale includes 16 items rated on a 5-point scale from 1 (*never or almost never*) to 5 (*always or almost always*), with higher scores indicating higher levels on each motive. The Cronbach’s alphas for the subscales were as follows: 0.69 for coping; 0.68 for conformity; 0.73 for enhancement; and 0.53 for social motive. The latter could not be used in this study as it did not achieve adequate internal consistency. 

#### 2.2.4. Vulnerable Narcissism

The Italian version of the Hypersensitive Narcissism Scale (HSNS; [37]; original version by [38]) was used to assess vulnerable narcissism. The HSNS includes 10 items (e.g., “My feelings are easily hurt by ridicule or by the hurtful remarks of others”) rated on a 5-point Likert scale that ranges from 1 (very uncharacteristic or untrue) to 5 (very characteristic or true). Higher total scores indicate greater levels of vulnerable narcissism. In the current study, Cronbach’s alpha was α = 0.65.

### 2.3. Data Analysis

Descriptive statistics and inter-correlations were calculated for all variables; to check for the normality of data distribution, we also evaluated skewness and kurtosis for each variable. Then the associations among variables were tested by computing structural equation models (SEM) with observed variables. These analyses were conducted with the Mplus software (version 8) [39]. As one assumption of mediation analysis is that the predictor and mediator variables must be significantly associated with the outcome variable, as a first step, we assessed all direct associations between the predictor variable (HSNS), the three mediators (coping, conformity, enhancement) and the five outcome variables (POSI, mood regulation, cognitive preoccupation, compulsive use, negative outcomes). Mediator and outcome variables were allowed to correlate as they are different dimensions of the same construct. Then we tested the indirect associations with a mediation model. Several indices of fit were used to assess the goodness of fit of the model: the root-mean-square error of approximation (RMSEA), the comparative fit index (CFI), and the standardized root-mean-square residual (SRMR). The cut-off criteria to determine adequate (CFI > 0.90, SRMR < 0.10, RMSEA < 0.08) and excellent fit (CFI > 0.95, SRMR < 0.08, RMSEA < 0.06) were those suggested by Hu and Bentler [40].

## 3. Results

Descriptive statistics and intercorrelations are reported in Table 2. Variables reported skewness and kurtosis values within the threshold of |2|, suggesting that the data distribution approximated normality [41]. Thus, the maximum likelihood estimator (ML) was used to estimate path coefficients. The results from the model, including all direct paths between predictor, mediator, and outcome variables, are reported in Figure 1 (fit indices are not reported as this first model was saturated). As expected, vulnerable narcissism was positively associated with all dimensions describing PSNSU (outcome variables). Similarly, vulnerable narcissism was positively associated with all motives for using SNSs (mediator variables). However, not all mediators and outcomes were significantly associated. Enhancement was the only motivation associated with all dimensions of PSNSU, while coping was only significant for mood regulation, cognitive preoccupation, compulsive use, and conformity were only significant for mood regulation and negative outcomes. 

The mediation model including all significant direct and indirect effects reported a good fit to our data (χ^2^ (3) = 5.37, *p* = 0.373, RMSEA = 0.015, 90% CI [0.000–0.077], CFI = 1.00, SRMR = 0.017). Coping partially mediated the association between vulnerable narcissism and mood regulation (b (SE) = 0.11 (0.03), *p* < 0.001), cognitive preoccupation (b (SE) = 0.05 (0.02), *p* = 0.001), compulsive use (b (SE) = 0.04 (0.02), *p* = 0.003). Conformity partially mediated the association between vulnerable narcissism and mood regulation (b (SE) = 0.03 (0.01), *p* = 0.015) and negative outcomes (b (SE) = 0.03 (0.01), *p* = 0.039). Enhancement partially mediated the association between vulnerable narcissism and POSI (b (SE) = 0.04 (0.01), *p* = 0.009), cognitive preoccupation (b (SE) = 0.07 (0.02), *p* = 0.002), compulsive use (b (SE) = 0.05 (0.02), *p* = 0.004), and negative outcomes (b (SE) = 0.06 (0.02), *p* = 0.003) but not mood regulation (b (SE) = 0.02 (0.01), *p* = 0.058).

## 4. Discussion

This study aimed to advance the understanding of the link between vulnerable narcissism and PSNSU symptoms among young adults. Our findings enrich the existing literature because they reveal the mediating roles of coping, conformity, and enhancement motives for SNSs use in this relationship. 

Consistently with our first hypothesis (H1), we found significant and positive associations between vulnerable narcissism and all the examined PSNSU symptoms. This finding is consistent with previous studies, which showed that SNSs provide ideal platforms to fulfill narcissistic needs [42], resulting in an increased risk of PSNSU, especially for vulnerable narcissists [14]. Furthermore, in line with a recent study [28], these associations are all of a similar magnitude (from weak to moderate), with the strongest direct association observed between vulnerable narcissism and POSI (β = 0.29). As vulnerable narcissism is characterized by ego-treat avoidance, people high in vulnerable narcissism may tend to prefer online social interactions over face-to-face ones as self-protection and preventive strategy [28,43]. In fact, specific features of SNSs (such as asynchronicity and absence of traditional social cues) might drive those people with difficulties in interpersonal relationships to carefully select and edit contents to be shared on SNSs for self-expression and (ideal) self-management [28]. However, the belief that one is safer and more comfortable in online social interactions does not actually lead to social need satisfaction but, on the contrary, expectations are often frustrated, thus exacerbating the levels of social anxiety and low social self-efficacy (e.g., [43]). Indeed, results indicated that vulnerable narcissists might also tend to use SNSs in order to deal with negative internal states and are likely to engage in deficient self-regulation in terms of worry about what happens online and compulsively using SNSs, thus experiencing negative consequences for daily life in terms of social and professional failure (e.g., [8,10]). When taken together, these results are in line with previous findings by Casale et al. [24], which showed that vulnerable narcissistic reported higher levels of all PSNSU symptoms compared to non-narcissistic and suggested that clinical interventions for these individuals should be targeted both at reducing addictive-like symptoms (e.g., mood modification) and providing strategies for achieving social needs alternative to POSI.

In accordance with our expectations (H2), the results showed that the link between vulnerable narcissism and mood regulation and cognitive preoccupation symptoms was partially mediated by coping and that the association between vulnerable narcissism and PSNSU mood regulation and negative outcomes symptoms was partially mediated by conformity. Our results are in line with the theory of compensatory Internet use [27], which states that vulnerable narcissists are more likely to develop PSNSU via motives for SNSs use with negative valence in an attempt to regulate unpleasant feelings associated with internal or external stressors [28]. In other words, people high in vulnerable narcissism might tend to reduce unwanted negative emotions (i.e., coping motives) and fears of not being liked by others or being excluded by a certain group of friends (i.e., conformity) by using SNSs. However, in turn, such need-seeking behaviors escalate in symptoms of PSNSU, in line with previous studies both on young adults [44] and adolescent SNSs users [32]. 

Interestingly, and partially contrary to our expectations, enhancement emerged as an important motive in the current model that partially mediated the association between vulnerable narcissism and all PSNSU symptoms, except for mood regulation. These findings suggest another possible mechanism to explain why vulnerable narcissists are at risk of PSNSU. Because of their low frustration tolerance [45], people with vulnerable narcissism may use SNSs to obtain immediate gratification [46]. In fact, SNSs may facilitate a gratification-seeking process (e.g., through controlling self-presentation [47]), as is the case in other forms of addictive behaviors (e.g., [48]). Furthermore, it should be noted that the enhancement subscale of the FMQ includes items that assess the usage of SNSs for their euphoric effects on mood (e.g., “How often do you use SNSs to experience a feeling of exaltation?”), which suggests a maladaptive nature of this dimension and consequent PSNSU [44]. Importantly, among the associations between enhancement and symptoms of PSNSU, the strongest was observed with cognitive preoccupation; that is, obsessively thinking about going on SNSs when offline. In other words, people with vulnerable narcissism engage in maladaptive SNSs use because of the increased immediate and positive emotions offered by SNSs but, in turn, are at major risk of developing tolerance-like symptoms in that they become preoccupied with the thought of going on SNSs when they have not been online for some time. Thus, this may represent another dysfunctional strategy utilized by vulnerable narcissists to regulate their dysphoric mood. An alternative explanation would be that, similarly to genuine addictions, PSNSU may involve a rewarding process in which positive affect facilitates SNSs use but, in the long run, decreases while negative affect increases, engendering a vicious circle. 

Overall, results suggested that the compensatory and gratification use of SNSs might be only apparent for vulnerable narcissists: on the one hand, indirect effects between narcissism and symptoms via the three motives indicate that individuals with vulnerable narcissism might feel and believe that SNSs are useful to compensate for their lack of social and emotional skills but such gratification seeking behavior results in problematic use; on the other hand, the direct effects between narcissism and symptoms suggest that other mechanisms (different from the motivational one) might be implicated in this process, such as attachment styles [13], perfectionism and metacognitive beliefs about the uncontrollability of thoughts and danger [44,49]. 

The present study has a number of limitations. First, given the cross-sectional design of the study, it is worth specifying that we used the term mediation only in the statistical sense. However, although it is not possible to rule out that other explanatory models may also fit the data, it is likely that a relatively stable personality characteristic (i.e., vulnerable narcissism) affects motivational and behavioral outcomes rather than the reverse. Further longitudinal studies are needed to disentangle the relationships among these variables. Second, we did not evaluate the different types of SNS used by participants nor examined specific forms of PSNSU (e.g., problematic Facebook use). In addition, the “social motive” subscale of the Facebook Motives Questionnaire could not be used in this study as it did not achieve adequate reliability. This may be due to several issues, including the small number of items and data completed online. Moreover, beyond self-reported SNSs use, objective data downloaded by SNSs profiles might give an interesting insight into actual activities preferred by users high in vulnerable narcissism [50,51]. Finally, our results may have been affected by third variables not examined here, such as insecure attachment, loneliness, or social anxiety. Further studies are needed with clinical samples of young adults, especially given the relevance of PSNSU in youth. 

## 5. Conclusions

Study limitations notwithstanding, we conclude that the relationship between vulnerable narcissism and PSNSU symptoms is partially mediated both by negative (i.e., coping and conformity) and positive (enhancement) motives for SNSs use. The study contributes to the body of knowledge by suggesting that individuals with vulnerable narcissism may develop PSNSU not only as a compensatory strategy to escape from offline problems but also as a result of a gratification-seeking process. From a theoretical point of view, these findings expand our current understanding of narcissistic vulnerability as a risk factor for PSNSU. From the practical point of view, they suggest that it is important for clinicians to evaluate and address which motives led individuals with vulnerable narcissism to PSNSU rather than focusing solely on behavioral addictive-like symptoms, such as the lack of control over one’s own use. In addition, educational programs aimed at reducing PSNSU should consider explicitly addressing hypersensitivity to rejection and disconfirmation as potential motivators of SNSs use. Suggestions for future research include long-term studies on larger samples. Furthermore, studies with clinical samples are needed to extend and strengthen our results and provide more definitive significance for public health. 

## Figures and Tables

**Figure 1 healthcare-10-01719-f001:**
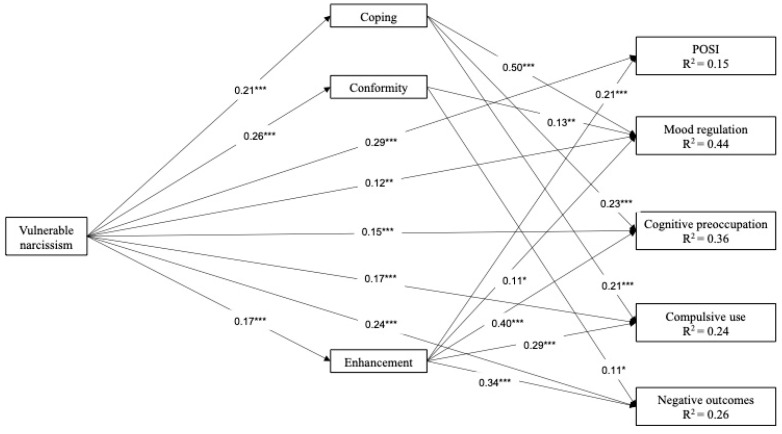
Mediation model with standardized effects. *Note.* Intercorrelations among mediator and outcome variables were included in the model and were significant at *p* < 0.001 but are not included in the figure for clarity of representation. *** *p* < 0.001; ** *p* < 0.01; * *p* < 0.05; *n* = 344; POSI = Preference for online social interactions.

**Table 1 healthcare-10-01719-t001:** Classification of motives for SNSs use based on the motivational model for problematic behaviors.

Source	Positive Valence	Negative Valence
**Internal**	Enhancement	Coping
**External**	Social	Conformity

**Table 2 healthcare-10-01719-t002:** Descriptive statistics and intercorrelations for all variables.

	*Variables*	1	2	3	4	5	6	7	8	9
1	Vulnerable narcissism	-	0.21 **	0.26 **	0.17 *	0.33 **	0.28 **	0.27 **	0.26 **	0.32 **
2	Coping		-	0.36 **	0.47 **	0.24 **	0.63 **	0.46 **	0.38 **	0.26 **
3	Conformity			-	0.48 **	0.21 **	0.40 **	0.37 **	0.29 **	0.35 **
4	Enhancement				-	0.26 **	0.42 **	0.53 **	0.41 **	0.43 **
5	POSI					-	0.36 **	0.39 **	0.25 **	0.35 **
6	Mood Regulation						-	0.53 **	0.38 **	0.35 **
7	Cognitive Preoccupation							-	0.68 **	0.54 **
8	Compulsive Use								-	0.59 **
9	Negative Outcomes									-
	*M*	27.85	2.99	2.27	2.31	1.94	4.46	2.53	3.88	1.92
	*SD*	5.64	0.79	0.74	0.76	1.14	1.96	1.18	1.86	1.21

*Notes*. * *p* < 0.01, ** *p* < 0.001; *n* = 344; POSI = Preference for online social interactions.

## Data Availability

The data that support the findings of this study are available from the corresponding author [A.M.] upon reasonable request.

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
