# Peer review of "Vulnerable Narcissism and Problematic Social Networking Sites Use: Focusing the Lens on Specific Motivations for Social Networking Sites Use"

_healthcare, 2022, doi:10.3390/healthcare10091719_

Round 1

Reviewer 1 Report

see attachment

Author Response

We thank the Reviewer for his/her careful review, which helped improve the manuscript. Our detailed, point-by-point responses to reviewer comments are given attached.

Reviewer 2 Report

Dear authors, congratulations on the study you present. The literature review is coherent and serves to lay the groundwork for the two hypotheses you put forward. They could have done more and could have come up with several explanatory models.

I found the reliability data of the scales surprising. Very low for what we are used to seeing in scientific studies. Even one of the variables could not be used because of its low reliability. It would be necessary to see why this happened.

Regarding the conclusions, it would be good to leave a specific section for them. The limitations and theoretical and practical implications are interesting.

Best regards.

Author Response

(The authors gave the same response as above.)
